# A Simple Narrative Review of Progress on the Processing and Utilization of Functional Rice

**DOI:** 10.3390/foods13233911

**Published:** 2024-12-03

**Authors:** Ziying Yi, Dagang Chen, Xinqiao Zhou, Jie Guo, Ke Chen, Chanjuan Ye, Chuanguang Liu, Juan Liu

**Affiliations:** 1Guangzhou Dublin International College of Life Sciences and Technology, South China Agricultural University, Guangzhou 510640, China; suby_yi1029@163.com; 2Rice Research Institute, Guangdong Academy of Agricultural Sciences, Guangzhou 510640, China; chendg@gdaas.cn (D.C.); 13428860986@139.com (X.Z.); guojie2503@126.com (J.G.); seekei@163.com (K.C.); chanjuanye@163.com (C.Y.); 3Guangdong Rice Engineering Laboratory, Rice Research Institute, Guangzhou 510640, China; 4Guangdong Key Laboratory of New Technology in Rice Breeding, Rice Research Institute, Guangzhou 510640, China; 5Key Laboratory of Genetics and Breeding of High Quality Rice in Southern China (Co-Construction by Ministry and Province), Ministry of Agriculture and Rural Affairs, Guangzhou 510640, China

**Keywords:** functional rice, colored rice, low gluten rice, high-resistant starch rice, micronutrient-rich rice, bioreactor rice

## Abstract

This paper aims to review the research progress on the processing and utilization of functional rice and explore its potential to enhance human health. The research progress in the processing and utilization of colored rice, low-protein rice, high-resistant starch rice, micronutrient-enriched rice, and bioreactor rice was systematically analyzed through a comprehensive literature review. The impact of various processing techniques on the retention of nutritional components in functional rice was also discussed. This study found that functional rice exhibits great potential in terms of nutritional value, health effects, and market demand. However, issues such as the loss of bioactive components during processing, the maintenance of specific agronomic traits, and market acceptance still need to be addressed. The research and development of functional rice are significant for enriching people’s dietary choices and addressing global malnutrition and chronic disease problems. Future efforts should focus on further optimizing processing techniques and utilizing genetic engineering and molecular breeding technologies to enhance the nutritional value and agronomic traits of functional rice, thus meeting market demands and health objectives.

## 1. Introduction

Rice is one of the world’s most vital staple crops, extensively cultivated across diverse regions, particularly in Southeast Asia, where it serves as the primary food source for millions. Providing a significant portion of daily caloric intake, rice supplies essential carbohydrates, protein, vitamins, and minerals necessary for maintaining human health [1]. According to the “WHO Guidelines 2023” [2] for some special groups, such as diabetes patients, it is recommended to appropriately reduce the intake of rice. However, it is essential to include a portion of staple foods in every meal. For weight management, it is recommended to control rice intake and avoid excessive energy intake. Given these pressing health concerns, enhancing the nutritional quality and functional properties of rice has become a shared priority across diverse fields, such as rice genetic breeding, scientific processing, and nutritional medicine. This literature review will explore recent research on innovative varieties of functional rice such as high γ-aminobutyric acid rice, resistant starch rice, low gluten rice, colored rice [3] and bioreactor rice, each offering unique health benefits.

In light of increasing public interest in health and nutrition, concepts like “nutrition-oriented agriculture” and “functional agriculture” have gained prominence. These approaches focus on producing food that not only sustains but also promotes health, addressing the growing demand for functional foods with preventive and therapeutic roles [4]. Functional rice, a specialized category of rice products, is an essential component of this movement. Historically, rice breeding has primarily aimed at traits like high yield and pest resistance, while research on functional rice has been relatively limited. However, recent societal changes all around the world have spurred a shift toward improving the nutritional quality of staple crops such as rice, aligning with national strategies like “Ecological Indicators” [5]. This paper aims to review the current status and advancements in functional rice research, emphasizing the importance of integrating nutritional enhancements into agricultural practices.

Functional rice is characterized by the presence of specific health-promoting components within its edible parts, contributing to nutritional balance, disease prevention, and overall well-being. In addition to essential macronutrients, functional rice varieties incorporate bioactive compounds that can positively influence physiological functions [6]. These compounds include antioxidants that combat oxidative stress, elements that enhance immune response, and substances that regulate metabolic processes. By leveraging these beneficial components, functional rice can significantly improve health outcomes and support the management of various health conditions [7]. The focus on functional foods, known for their natural, safe, and effective properties, positions functional rice as a key area of development within China’s food industry for the 21st century. Promoting and commercializing functional rice not only aims to enhance the nutritional status of rice-dependent populations but also contributes to agricultural transformation and rural revitalization efforts [8]. Functional rice is not only for the “patient” and for the auxiliary treatment of disease, but it also has a good disease prevention effect that can be suitable for all people. However, the existing consumption habits of rice as a staple food and the limitation of the public cognition of functional rice make the vast majority of people misunderstand the role of functional rice and think that it is necessary to choose functional rice instead of conventional rice only when they are sick. Therefore, correct and effective publicity and public opinion guidance are indispensable. This paper synthesizes the existing literature on functional rice, elucidates the mechanisms behind its health benefits, and identifies challenges and future directions for research in this promising field. The classification of functional rice is shown in Figure 1.

## 2. Colored Rice

### 2.1. Processing and Utilization of Colored Rice

The growing demand for nutritious and health-promoting foods has renewed interest in colored rice varieties due to their unique nutritional profiles and health benefits, especially in Southeast Asia. Compared to traditional white rice, colored rice varieties such as black rice, red rice, and purple rice [9] are rich in antioxidants, vitamins, minerals, and flavonoids [10], offering several health advantages. For instance, black rice contains abundant anthocyanins, which provide powerful antioxidant properties [11], supporting vision health, boosting immunity, and reducing cardiovascular disease risk [12]. Red rice is a rich source of iron and vitamin B6, which helps prevent iron deficiency anemia [13] and stabilizes blood sugar levels. Radiation-induced mutagenesis is an effective technique for developing new colored rice varieties. In one study, researchers used the γ-ray radiation method with the 9311 variety as a parent to produce a darker grain with increased anthocyanin content. The resulting mutants showed enhanced levels of fat, amylopectin, flavonoids, and anthocyanins [14].

Colored rice grains are rich in bioactive compounds and essential nutrients. However, traditional rice processing methods often fail to preserve these valuable components, making it crucial to develop and optimize new techniques to retain nutritional value. Current research primarily focuses on three types of processing methods: physical, biological, and chemical treatments. Physical and chemical technologies are collectively referred to as non-biological processing technologies to treat grains, such as cooking, baking, high-pressure and low-temperature plasma, etc., with simple operation, high production efficiency, and low energy consumption. Physical treatments aim to enhance processing efficiency and product quality by modifying the physical properties of the grains. Heat treatments, such as boiling and roasting, promote starch gelatinization, making the grains edible. Roasting, in particular, improves the degradability, antioxidant capacity, storage stability, and functionality of brown rice flour [15]. It can soften the structure of the brown rice bran layer, weaken its hardness, and shorten the cooking time. Extrusion technology modifies the structure of colored rice, deactivating lipase to extend shelf life, and increasing the bioavailability of nutrients. For instance, extrusion processing enhances the total phenolic and anthocyanin content in black rice bran, significantly improving its antioxidant activity [16]. High-pressure processing is another effective method, sterilizing the grains while preserving their pigments and nutritional components [17]. Ultra-high pressure treatment can effectively inhibit fatty acid rout during brown rice storage, and the contents of phenolic, flavonoid, and other antioxidant active substances in brown rice are positively correlated with pressure [18]. Biological treatments, including fermentation, germination, and enzymatic processing, offer gentler methods for improving grain quality. These processes decompose dietary fiber, protein, starch, and other macromolecules in brown rice, make the internal structure of brown rice loose, reduce the resistance of water in and out of brown rice, promote the precipitation of water-soluble starch, and improve the viscosity of brown rice. Fermentation using probiotics, such as lactic acid bacteria, increases antioxidant properties and boosts the bioavailability of functional compounds [19]. Germination promotes the accumulation of γ-aminobutyric acid (GABA) and polyphenols, while enzymatic treatments with cellulase improve the digestibility and cooking quality of the rice [20]. Chemical treatments focus on preserving bioactive compounds and improving grain stability during storage. Techniques such as radiofrequency, ultrasound, and microwave heating reduce microbial growth and nutrient loss. Ultrasound-assisted treatment stabilizes pigments and enhances the retention of anthocyanins, while microwave heating provides rapid, uniform heat that minimizes nutrient degradation. When combined with physical methods, chemical treatments can further enhance nutritional preservation and prolong shelf life [21]. In addition to changing hardness and viscosity, the texture characteristics of brown rice taste are also affected by elasticity, chewability, roughness, and cohesiveness.

Optimizing these processing methods ensures that colored rice retains its nutritional benefits while improving consumer acceptance. As research advances, the development of innovative technologies will support the large-scale production of high-quality colored rice products, offering new opportunities to meet market demand for healthy, functional foods.

### 2.2. Challenges and Solutions for Colored Rice

Traditional processing methods often fail to preserve bioactive compounds in colored rice, particularly antioxidants like anthocyanins, which can degrade under high temperatures and light. To address this, milder processing techniques, such as vacuum freeze-drying and microwave-assisted drying, should be developed to enhance food safety while maximizing nutritional retention. Research on the health benefits and environmental impacts of colored rice remains limited. Future studies should explore the interactions between anthocyanins in black rice and gut microbiota, as well as the metabolic pathways of polyphenolic compounds to reveal their antioxidant effects [22]. Additionally, applying inorganic fertilizers can significantly boost the yield and anthocyanin content in purple rice, while magnesium supplementation enhances its metabolic processes. Market acceptance is also crucial. Public awareness of colored rice’s health benefits is limited, necessitating educational campaigns to improve consumer understanding and acceptance. The unfamiliar appearance and higher prices of colored rice hinder its popularity, so effective marketing strategies are needed to increase consumer interest. The processing, utilization, and challenges of functional rice are shown in Figure 2.

## 3. Low-Glutelin Rice

### 3.1. Processing and Utilization of Low-Glutelin Rice

The rise in chronic diseases such as kidney disorders, diabetes, and obesity has led to increased interest in developing low-glutelin rice as a dietary solution. Rice proteins can be classified into four types based on solubility: albumins (water-soluble), globulins (salt-soluble), prolamins (alcohol-soluble), and glutelins (acid or alkali-soluble). Glutelins comprise about 80% of rice protein content and are known for their excellent digestibility and antioxidant properties [23]. However, researchers in Japan and China have found that individuals with specific health conditions may need to limit protein intake to alleviate metabolic burdens. Low-glutelin rice varieties have been developed to cater to these needs, with water-soluble protein content ranging from 3.1% to 4.0% [24]. This reduction supports patients with diabetes and kidney disease by minimizing glucose conversion and easing protein metabolism. Researchers have successfully created several low-glutelin rice varieties through a combination of traditional breeding and biotechnological methods. Notably, the use of CRISPR/Cas9 gene-editing technology has shown significant potential in enhancing the quality of these rice varieties [25].

Processing techniques for low-glutelin rice are crucial in preserving its unique nutritional profile. Traditional milling methods often result in significant nutrient loss, particularly in polished rice. Therefore, researchers are exploring innovative techniques that minimize nutrient loss while enhancing the quality of the grains. Physical processing methods [26], such as washing and high-pressure sterilization, play a significant role in maintaining the quality of low-glutelin rice. High-pressure treatment enhances the solubility of proteins and increases nutrient accessibility. Additionally, washing rice prior to cooking can help remove surface contaminants and improve the overall safety and quality of the product. The solubility, emulsification, and foamability of rice protein were improved by heat treatment, high-pressure microjet treatment, and freezing grinding treatment [27,28]. Extrusion is another key processing method that reduces moisture content and protein levels while breaking down proteins into smaller peptides and amino acids. This technique not only improves the digestibility of the rice but also enhances its functional properties. Research has shown that combining extrusion with enzymatic treatments can further improve solubility, achieving protein reduction rates of up to 27.47% [27]. For instance, enzymatic fermentation using lactic acid bacteria has proven effective in degrading proteins and improving the quality of rice, offering promising applications for low-protein dietary products [29].

Modern breeding efforts increasingly integrate molecular markers with traditional techniques to create low-glutelin rice varieties that also exhibit desirable agronomic traits. For example, researchers at the Shanghai Academy of Agricultural Sciences have developed new strains by crossbreeding LGC-1 with high-resistant starch varieties, yielding functional rice that boasts both low protein content and improved starch quality [30]. The Guangdong Academy of Agricultural Sciences has similarly developed hybrids that combine low-glutelin and disease-resistant characteristics through targeted breeding and backcrossing [31].

### 3.2. Challenges and Solutions for Low-Glutelin Rice

Despite advancements, challenges remain in balancing agronomic performance, taste, and nutritional functionality in low-glutelin rice. While gene-editing technologies like CRISPR/Cas9 have accelerated breeding efforts, focusing on specific traits may overlook other critical agronomic factors. Additionally, chemical modification techniques, although effective, risk compromising the nutritional integrity of proteins and may leave toxic residues. Research is ongoing to explore milder processing methods, such as hydrothermal treatments, which improve functional properties without damaging nutritional quality [32]. Addressing the challenges associated with the processing, market acceptance, and agronomic traits of low-glutelin rice is essential for its successful commercialization. Ongoing research and development should prioritize optimizing processing methods [33], increasing consumer awareness, and enhancing breeding strategies to ensure low-glutelin rice meets the growing demand for healthy food options.

## 4. High-Resistant Starch Rice

### 4.1. Processing Techniques for High-Resistant Starch Rice

High-resistant starch (RS) refers to the portion of starch that is not digested in the small intestine and is fermented in the colon [28], producing short-chain fatty acids beneficial for gut health. High-resistant starch rice is characterized by a low glycemic index (GI), which can effectively lower postprandial blood sugar levels and offers multiple health benefits, including antihypertensive and antioxidant properties [34]. The increase in amylose content significantly enhances the RS levels in rice grains, providing a promising avenue for functional food development. Research has shown that growing high-resistant starch rice varieties during early seasons is advantageous for slowing starch digestion rates [35]. These varieties are being developed through advanced breeding techniques and molecular genetics. For instance, studies have identified key regulatory pathways and potential genes involved in RS synthesis, which will facilitate targeted breeding efforts aimed at improving RS content in rice [36].

The development of high-resistant starch rice is complemented by innovative processing techniques that enhance its nutritional profile and functional properties, especially in Southeast Asia. Utilizing CRISPR/Cas9 technology to edit starch branching enzyme genes has resulted in rice varieties with altered starch structures, leading to increased RS content [37]. Harvesting, extraction, and purification of the proteins produced in high-resistant starch rice require optimized methods to ensure that high-quality high-resistant starch rice production is more effective in promoting mineral absorption. High-pressure sterilization and cooling treatments have shown promise in enhancing RS levels while maintaining grain quality and safety. Emerging methods, such as ultrasonic treatment and microwave processing, are being explored for their ability to modify starch structures and increase RS yield [38]. These methods are known for their efficiency and minimal nutrient loss compared to traditional processing techniques. For instance, combining physical and enzymatic treatments [39] can significantly improve the nutritional properties of high-resistant starch rice, resulting in enhanced functional food products [40]. Moreover, ensuring that high-resistant starch rice retains its desirable sensory attributes, such as texture and flavor, is critical for consumer acceptance. Research should also focus on developing guidelines for processing high-resistant starch rice that balances nutritional enhancements with sensory qualities. After the rice was treated with a GABA enrichment device, the content of reduced sugar and total starch decreased, while the content of resistant starch increased, making the rice products after treatment more suitable for diabetic patients [41].

RS nanoparticles are stable colloidal dispersions with good colloidal stability, which can be used in acidic and neutral food dispersions, such as carbonated beverages, non-carbonated beverages, fruit juices, condiments, sauces, etc., and can be widely used in the development of beverages rich in dietary fiber [42].

### 4.2. Challenges and Solutions for High-Resistant Starch Rice

Despite the promising developments in high-resistant starch rice, several challenges remain. One of the main issues is the limited understanding of the genetic and environmental factors influencing RS synthesis. This knowledge gap complicates breeding efforts aimed at developing varieties with consistently high RS content. Furthermore, the textural properties of high-resistant starch rice, such as hardness and cooking quality, can deter consumer acceptance. Therefore, balancing RS content with sensory attributes is crucial for market viability. Future research should focus on elucidating the underlying mechanisms of RS synthesis, optimizing breeding strategies, and developing processing techniques that enhance both nutritional and sensory qualities [35]. In summary, addressing the challenges associated with high-resistant starch rice through targeted research and development will be essential for its successful commercialization. By enhancing breeding methods and processing techniques [43], the potential of high-resistant starch rice to meet the growing consumer demand for healthy food options can be fully realized.

## 5. Micronutrient-Rich Rice

### 5.1. Processing Techniques for Micronutrient-Rich Rice

Micronutrient-rich rice is designed to address the nutritional deficiencies prevalent in many populations, particularly in regions where rice is a staple food, such as Japan, China, and Thailand. Key micronutrients often lacking in diets include iron, zinc, and vitamins A and C [3]. The development of rice varieties fortified with these essential nutrients is critical for improving public health and reducing malnutrition. Recent research has focused on biofortifying rice with micronutrients through various strategies, including traditional breeding methods, genetic engineering, and agronomic practices. Genetic engineering has enabled scientists to introduce and express genes responsible for micronutrient uptake and accumulation directly in rice plants. For instance, researchers have successfully developed transgenic rice varieties that contain elevated levels of iron and zinc, providing a more nutrient-dense food source for consumers [44].

Processing methods are vital for preserving and enhancing the micronutrient content of rice. Traditional milling processes often result in the loss of essential nutrients, particularly in polished rice. Therefore, researchers are investigating techniques that minimize nutrient loss while improving grain quality. One effective approach is parboiling, which involves soaking rice in water and then steaming it. This method not only improves nutrient absorption but also enhances the bioavailability of minerals, making them more accessible to the body [45]. Additionally, fortification techniques, such as coating rice grains with micronutrients, are being researched to enrich rice further before consumption. Studies have demonstrated that pre-milling treatments can significantly increase the micronutrient content in rice. For example, soaking rice in solutions enriched with vitamins and minerals can lead to higher retention rates during cooking [46]. This approach is particularly beneficial for enhancing the iron and zinc content in rice, addressing common deficiencies in populations reliant on rice as a staple food.

Rice germ has rich nutritional value and high plasticity, its protein, high-quality fat, minerals, vitamin B1 and vitamin E and other nutrients can be added to flour for making biscuits, bread, and other products, and can also be directly used in the production of rice germ oatmeal, rice germ beverage, rice germ yogurt and other functional foods [47]. Educating consumers about the benefits of processing techniques that enhance micronutrient content is essential for increasing the acceptance of fortified rice varieties. Effective marketing strategies that highlight the health advantages of micronutrient-rich rice can also play a significant role in promoting these products.

### 5.2. Challenges and Solutions for Micronutrient-Rich Rice

Characteristic agriculture rich in selenium, iron, and zinc has gradually become a research hotspot, but most of the existing research focuses on the analysis of a single element in rice, while the research on trace elements in different rice varieties and their dietary intake is limited [48]. These elements such as selenium, iron, and zinc are essential for the body’s physiological function, but too much or too little can cause health problems, so exposure to toxic elements such as arsenic, cadmium, nickel, and lead must be minimized [49]. Further studies indicate that deficiencies in elements such as cobalt (Co), iron (Fe), and manganese (Mn) can lead to anemia and growth inhibition, while excessive concentrations of these elements may result in neurological disorders, hemochromatosis, cognitive impairments, and other health issues. In some areas of China, the lack of medium and trace elements in farmland has been supplemented according to the soil condition, which not only improves soil fertility and fertilizer utilization efficiency, but also significantly improves the yield and quality of rice. In summary, advancing the research, processing technologies, and market acceptance of micronutrient-rich rice is vital for its successful implementation. By addressing these challenges and enhancing the nutritional value of rice, we can significantly contribute to global efforts in combating malnutrition.

## 6. Bioreactor Rice

### 6.1. Processing Techniques for Bioreactor Rice

Bioreactor rice is engineered to produce pharmaceutical proteins and bioactive compounds, which has a broad application, making it a promising tool in the field of biomedicine. To enhance the immune response of the vaccine, transgenic rice expressing the novel coronavirus S1 protein was successfully developed, and its immunogenicity was evaluated in combination with flagellin protein (FliC) [50]. In recent years, microbial heterologous synthesis of melatonin has emerged as a research hotspot due to its cost-effectiveness and sustainability. Additionally, biotechnological advancements in rice have demonstrated significant potential to enhance the efficiency of melatonin synthesis [51]. This approach offers a novel strategy for the development of vaccines against the novel coronavirus. By utilizing rice as a biofactory, researchers can harness its natural growth and reproductive systems to express complex proteins, such as vaccines, therapeutic enzymes, and antibodies [52]. Recent advancements in genetic engineering have enabled the development of transgenic rice varieties capable of synthesizing therapeutic proteins. For instance, studies have demonstrated that rice can effectively produce human insulin and other medical proteins with high purity and yield [53]. Furthermore, the use of bioreactor rice for vaccine production is gaining traction, particularly in developing oral vaccines that are stable and easy to administer [54].

Processing techniques are critical for maximizing the efficiency and yield of bioreactor rice. The extraction and purification of the pharmaceutical proteins produced in rice grains require optimized methods to ensure high-quality outcomes. A common approach involves using protease inhibitors during the extraction process to prevent the degradation of sensitive proteins [55]. Advanced purification techniques, such as chromatography and filtration, are employed to isolate the desired proteins while removing impurities effectively. Researchers are also exploring innovative extraction methods, including ultrasonic and microwave-assisted extraction, which may enhance yield and reduce processing time. Plant hormones also had significant effects on melatonin synthesis. Exogenous gibberellin treatment could significantly induce melatonin synthesis in rice seedlings, while abscisic acid inhibited it [56]. These advanced techniques have shown the potential to increase extraction efficiency while preserving the integrity of the target proteins.

The effectiveness of these processing methods can vary depending on the specific protein being targeted and the cultivation conditions of the rice. Therefore, ongoing research is necessary to develop standardized protocols that optimize the extraction and purification of proteins from bioreactor rice.

### 6.2. Challenges and Solutions for Bioreactor Rice

Despite its potential, the development of bioreactor rice faces several challenges. One of the primary concerns is the regulatory framework surrounding genetically modified organisms (GMOs). The complexity of the regulations can delay the approval and commercialization of bioreactor rice products. To address these challenges, it is essential to engage with regulatory agencies early in the research process to ensure compliance with safety and environmental standards. Additionally, public education campaigns that highlight the benefits of bioreactor rice, such as its role in sustainable medicine and food security, can help mitigate public resistance to GMOs [57]. Another challenge is ensuring that the cultivation of bioreactor rice does not adversely influence biodiversity or the environment. Implementing sustainable agricultural practices and conducting thorough environmental impact assessments can help maintain ecological balance while promoting the cultivation of bioreactor rice. In conclusion, advancing the research, processing technologies, and market acceptance of bioreactor rice is crucial for its successful implementation. By addressing regulatory and environmental challenges, we can harness the potential of bioreactor rice to contribute to advancements in medicine and public health [58].

## 7. Conclusions

As consumers become increasingly aware of the impact of diet on health, they actively seek foods with specific health benefits, thereby driving demand for functional foods. However, fully realizing the potential of functional foods presents a considerable challenge. Long-term consumption of refined rice alone can lead to micronutrient deficiencies, while nutrients such as resistant starch, γ-aminobutyric acid, folic acid, β-carotene, astaxanthin, anthocyanins, iron, zinc, selenium, and short peptides found in nutrient-enhanced rice play crucial roles in boosting immunity and preventing chronic diseases. Advancing the nutritional rice industry not only promotes public health but also enhances socio-economic outcomes. Future directions in functional rice breeding will shift from single-function traits to multifunctional ones (e.g., health, social, and economic aspects), progressing from a health-focused approach to one combining health promotion and therapeutic support. Research in functional rice will rely on close interdisciplinary integration across genetics, physiology, nutrition, and medicine, with new ideas, methods, and technologies emerging from these intersections, laying the foundation for continual breakthroughs in this field.

## Figures and Tables

**Figure 1 foods-13-03911-f001:**
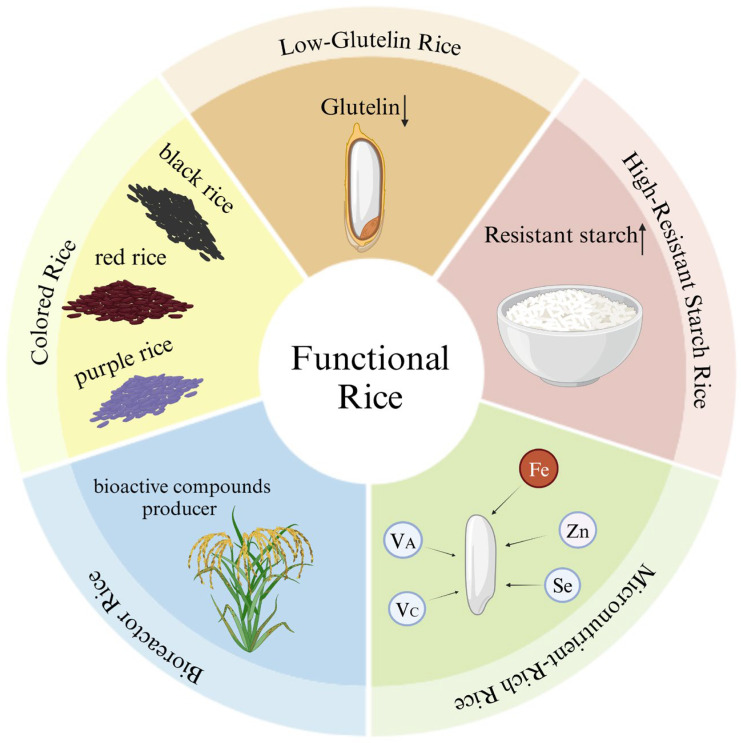
The classification of functional rice.

**Figure 2 foods-13-03911-f002:**
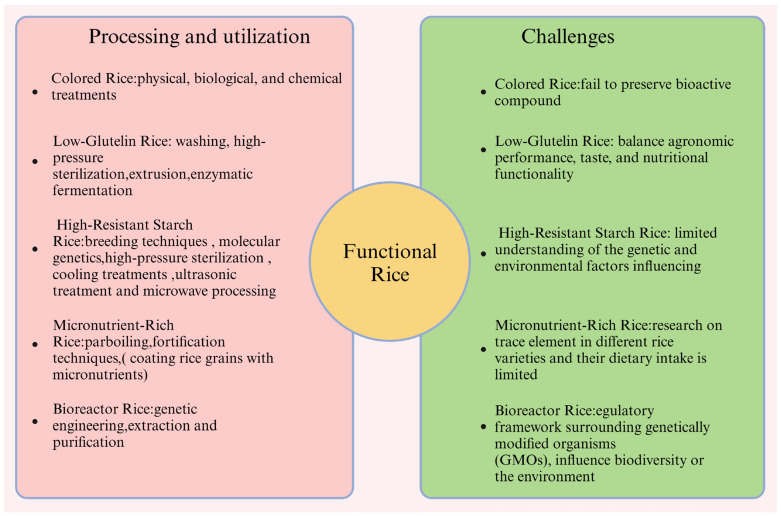
Processing, utilization, and challenges of functional rice.

## Data Availability

No new data were created or analyzed in this study. Data sharing is not applicable to this article.

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
