# Peer review of "A Simple Narrative Review of Progress on the Processing and Utilization of Functional Rice"

_foods, 2024, doi:10.3390/foods13233911_

Round 1
Reviewer 1 Report
Comments and Suggestions for Authors
The authors present an article named “Research progress on the processing and utilization of functional rice” in which a narrative or literature review regarding aspects of specific rice, such as processing techniques and biofunctional activities are approached. The English is good. The methodoloy could not be evaluated since there is no description of a review methodology. One of my major points for review is the lack of methodology either the authors insert their methodology or let it more explicit in the title that this a simple narrative review. Another point is the visual aid for data visualization, there is no graph or tables. In this article the authors could easily have made one table per section and maybe an overall graph to compare the data between the types of functional rice. Also, some in text parts seem to be either to locally focused (China) or somewhat vague; Otherwise, the article is simple, but relevant.
The authors should let it clear in the article’s title or in the beginning of the introduction, that this is a literature review, because there is not a methodology section describing the review’s approach
L 33-36 I suggest that the authors look for more globally related guidelines instead of only China’s guidelines, to extrapolate the discussion worldwide
L48 Again the program “Healthy China” is not common sense globally, please use a worldwide example or explain the country-local references, in further similar cases
L85-94 The authors can organize the information on processing methods for better readbility
L130 The use of tables showing the processing techniques in Section 3 would also improve the readability, mostly with the enzimatic and physical methods, showing values in nutritional loss as the authors comment on
L204 “high-quality outcomes” the parameter is vague, please explain clearly what the quality parameter are the authors writing about, and which outcomes are they referring to, this is valid for other instances throughout the text that I didn’t comment here
L337-338 “will shift from single-function traits to multifunctional ones” The authors can give some examples of the multifunctional traits specifically related to functional rice, to reduce vagueness
Author Response
Many thanks for the valuable suggestions or comments on our manuscript from the reviewers. As suggested, the manuscript has been examined and revised carefully while the textual and typographical errors have been corrected.
Reviewer #1:
The reviewers’ comments:
The authors present an article named “Research progress on the processing and utilization of functional rice” in which a narrative or literature review regarding aspects of specific rice, such as processing techniques and biofunctional activities are approached. The English is good. The methodoloy could not be evaluated since there is no description of a review methodology. One of my major points for review is the lack of methodology either the authors insert their methodology or let it more explicit in the title that this a simple narrative review. Another point is the visual aid for data visualization, there is no graph or tables. In this article the authors could easily have made one table per section and maybe an overall graph to compare the data between the types of functional rice. Also, some in text parts seem to be either to locally focused (China) or somewhat vague; Otherwise, the article is simple, but relevant.
The authors’ response:
Thanks for your professional comments, we have changed the title of this article to a simple review because we do not have a professional methodology. More charts on functional rice are also provided. Finally, there are some vague concepts of the region also added to the Southeast Asian countries, which rely on rice as the staple food
The reviewers’ comments:
The authors should let it clear in the article’s title or in the beginning of the introduction, that this is a literature review, because there is not a methodology section describing the review’s approach.
The authors’ response:
Thanks for your advice, we have made the beginning of the introduction clearer and added that this is a literature review.
The reviewers’ comments:
L 33-36 I suggest that the authors look for more globally related guidelines instead of only China’s guidelines, to extrapolate the discussion worldwide.
The authors’ response:
Thanks for your suggestion. After further understanding, I replace China's guidelines with WHO guidelines, which is more authoritative.
The reviewers’ comments:
L48 Again the program “Healthy China” is not common sense globally, please use a worldwide example or explain the country-local references, in further similar cases
The authors’ response:
Thanks for your valuable advice, I have changed Healthy China, a periodical that is too strong ina specific region, to a more global periodical Ecological Indicators.
The reviewers’ comments:
L85-94 The authors can organize the information on processing methods for better readbility
The authors’ response:
Thank you very much for receiving your valuable advice. But the reliable treatment of colored rice has been summarized into three treatments: physical, biological, and chemical.
The reviewers’ comments:
L130 The use of tables showing the processing techniques in Section 3 would also improve the readability, mostly with the enzimatic and physical methods, showing values in nutritional loss as the authors comment on
The authors’ response:
Thanks for the careful comments, we have revised it.
The reviewers’ comments:
L204 “high-quality outcomes” the parameter is vague, please explain clearly what the quality parameter are the authors writing about, and which outcomes are they referring to, this is valid for other instances throughout the text that I didn’t comment here
The authors’ response:
Thank you for your careful comment. We have detailed high quality outcomes as high quality high resistant starch rice production is more effective in promoting mineral absorption. More detailed explanation to high quality results.
The reviewers’ comments:
L337-338 “will shift from single-function traits to multifunctional ones” The authors can give some examples of the multifunctional traits specifically related to functional rice, to reduce vagueness
The authors’ response:
Thank you for your very careful comments. The versatility of rice can be reflected in the restoration of human health, the contribution to social development and the promotion of economic development. These factors are summarized in the article.
Reviewer 2 Report
Comments and Suggestions for Authors
Rice is extremely important for the economy and is the basis of the diet of the population of many countries, so the research of new strategies on improving agricultural production and nutritional quality of the grain is of paramount importance.
1- It is important to describe how the literature review was carried out, even if it is a narrative review. It is importante to describe Suitable studies were found through the use of the electronic search systems (PubMed, Google Scholar, and Scopus ???) for the literature search.
2- It is importante to describe searched the bibliographies to identify relevant studies and reviews.
3-The search in the databases was performed by combining search terms?
4-Are there studies on the acceptance of these new types of rice? It would be important to add.
5- Line 140: "This reduction supports patients with diabetes and kidney disease by minimizing glucose conversion and easing protein metabolism".Explain how minimizing glucose conversion.
Author Response
Many thanks for the valuable suggestions or comments on our manuscript from the reviewers. As suggested, the manuscript has been examined and revised carefully while the textual and typographical errors have been corrected.
Reviewer #2:
The reviewers’ comments:
Rice is extremely important for the economy and is the basis of the diet of the population of many countries, so the research of new strategies on improving agricultural production and nutritional quality of the grain is of paramount importance.
1- It is important to describe how the literature review was carried out, even if it is a narrative review. It is importante to describe Suitable studies were found through the use of the electronic search systems (PubMed, Google Scholar, and Scopus ???) for the literature search.
The authors’ response:
Thank you for your objective comments. Rice is really a very important crop for the economy, especially in Southeast Asia where rice is a staple food. For this literature review, in order to ensure its rigor and authority, we searched many authoritative academic websites such as PubMed, Google Scholar and Scopu to find the most suitable articles.
The reviewers’ comments:
2- It is importante to describe searched the bibliographies to identify relevant studies and reviews.
The authors’ response:
Thank you for your careful comments. We strongly agree that correct literature is very important for literature review. Therefore, for the compilation of this paper, we spent a lot of time searching for relevant literature, reading and understanding each literature in detail, and finally selected the latest literature that is more suitable for the theme, more authoritative and published in recent years.
The reviewers’ comments:
3-The search in the databases was performed by combining search terms?
The authors’ response:
Thank you for your academic question. In this literature review, we have identified five types of functional rice when searching relevant literature, and conducted keyword search according to different cultivation and processing methods of each functional rice and different contents.
The reviewers’ comments:
4-Are there studies on the acceptance of these new types of rice? It would be important to add.
The authors’ response:
Thanks for the careful comments, we have added “Functional rice is not only for the "patient" and for the auxiliary treatment of disease, it also has a good disease prevention effect that can be suitable for all people. However, the existing consumption habits of rice as a staple food and the limitation of the public cognition of functional rice make the vast majority of people misunderstand the role of functional rice and think that it is necessary to choose functional rice instead of conventional rice only when they are sick. Therefore, correct and effective publicity and public opinion guidance are indispensable.”
The reviewers’ comments:
5- Line 140: "This reduction supports patients with diabetes and kidney disease by minimizing glucose conversion and easing protein metabolism". Explain how minimizing glucose conversion.
The authors’ response:
Thanks for the careful comments, we have revised it to “This reduction supports patients with kidney disease by easing protein metabolism”.
Reviewer 3 Report
Comments and Suggestions for Authors
Comments on the manuscript entitled " Research progress on the processing and utilization of functional rice " by Yi et al.
The authors reviewed processing, utilization, challenges, and solutions for colored rice, low-glutelin rice, high-resistant starch rice, micronutrient-rich rice, and bioreactor rice. Overall, the manuscript is well-written and analysed. However, minor corrections should be made before its acceptance:
1. Please rewrite the text on the processing of colored rice (lines 83-109) to include a comparison of physical and chemical methods. Are chemical methods a better option for preserving the bioactive profile of rice? Also, please clarify how physical methods affect the physical properties of rice.
2. Please consider ultra-high pressure as s processing methods for colored rice as well.

Author Response
Many thanks for the valuable suggestions or comments on our manuscript from the reviewers. As suggested, the manuscript has been examined and revised carefully while the textual and typographical errors have been corrected.
Reviewer #3:
The reviewers’ comments:
- Please rewrite the text on the processing of colored rice (lines 83-109) to include a comparison of physical and chemical methods. Are chemical methods a better option for preserving the bioactive profile of rice? Also, please clarify how physical methods affect the physical properties of rice.
The authors’ response:
Thank you for your valuable comments. Most processing methods will combine physical and chemical processing methods, so physical and chemical technology is collectively referred to as non-biological processing technology to treat grains, such as cooking, baking, high pressure and low temperature plasma, etc., with simple operation, high production efficiency, low energy consumption characteristics. In addition to hardness and viscosity, the texture properties of brown rice affect the taste of elasticity, chewability, roughness, cohesion and so on.
The reviewers’ comments:
- Please consider ultra-high pressure as s processing methods for colored rice as well.
The authors’ response:
Thank you very much for your valuable comments. Ultrahigh pressure treatment is indeed a very important method. We added ultrahigh pressure treatment which can effectively inhibit fatty acid rout during brown rice storage.
Reviewer 4 Report
Comments and Suggestions for Authors The paper “Research progress on the processing and utilization of functional rice” looks like a collection of introduction related to five types of functional rice, e.g colored Rice, low-glutelin rice, high-resistant starch rice, micronutrient-rich Rice and bioreactor rice. I have to confess that, reading your paper, I found less useful information. My major concern is related to the lack of tables (with properties), figures (materials, techniques, methods of analysis and results), particular examples and a discussion about the correlation between the chosen materials (see my point 6 and 7 for particular suggestions). My comments are into a limited number, but mainly this is due to the limited information to be analyzed. Therefore, my recommendation for editor was that the paper should not be accepted in this summary form. Please take into consideration my concerns, include useful information (see above), compare the rice among them, rewritten and the resubmitted the paper. Some of my other observations are listed below: 1. Abstract: “Through a literature review, the research progress of colored rice, low-glutelin rice, high-resistant starch rice, micronutrient-rich rice, and bioreactor rice was analyzed.” In which sense the research progress? Please be more specific and revise this sentence! 2. Introduction: “but each meal should still eat some staple food.” Please rephrase! 3. Please rephrase “Given these pressing health concerns, enhancing the nutritional quality and functional properties of food is a shared priority across various fields such as crop genetics, food science, and nutritional medicine”, to be more clear! 4. From introduction and keywords it is not clear that the colored rice; low gluten rice; high resistant starch rice; micronutrient-rich rice; are various types of functional rice or are different types of rice and functional rice is another type of rice! Please clarify! 5. Keywords: “bioreactor” or “bioreactor rice”? Are two different concepts! 6. You mentioned “These elements such as selenium, iron and zinc are essential for the body's physiological function, but too much or too little can cause health problems, so exposure to toxic elements such as arsenic, cadmium, nickel and lead must be minimized [45].” My expectation from a review is to provide these limits (too much or too little) and to discuss about the content of these elements in the chosen materials (here the rice). This would be valuable information.You mentioned “Bioreactor rice is engineered to produce pharmaceutical proteins and bioactive compounds, making it a promising tool in the field of biomedicine.” This is a potential, interesting information. The readers would like to find out some examples of application. Else this review is maintained at a general discussion.
Comments on the Quality of English LanguageIn places (especially in abstract and introduction) must be improved.
Author Response
Many thanks for the valuable suggestions or comments on our manuscript from the reviewers. As suggested, the manuscript has been examined and revised carefully while the textual and typographical errors have been corrected.
Reviewer #4:
The reviewers’ comments:
The paper “Research progress on the processing and utilization of functional rice” looks like a collection of introduction related to five types of functional rice, e.g colored Rice, low-glutelin rice, high-resistant starch rice, micronutrient-rich Rice and bioreactor rice. I have to confess that, reading your paper, I found less useful information. My major concern is related to the lack of tables (with properties), figures (materials, techniques, methods of analysis and results), particular examples and a discussion about the correlation between the chosen materials (see my point 6 and 7 for particular suggestions). My comments are into a limited number, but mainly this is due to the limited information to be analyzed. Therefore, my recommendation for editor was that the paper should not be accepted in this summary form. Please take into consideration my concerns, include useful information (see above), compare the rice among them, rewritten and the resubmitted the paper.
The authors’ response:
Thank you very much for your patient and professional comments. We have supplemented the chart, and we will listen to all your constructive suggestions with an open mind and make corrections. Thanks again for your careful advice.
The reviewers’ comments:
- Abstract: “Through a literature review, the research progress of colored rice, low-glutelin rice, high-resistant starch rice, micronutrient-rich rice, and bioreactor rice was analyzed.” In which sense the research progress? Please be more specific and revise this sentence!
The authors’ response:
Thank you for your careful comments. We changed this sentence to review the research progress in the processing and utilization of five functional rice varieties, which became more detailed and more in line with the theme of our article.
The reviewers’ comments:
- Introduction: “but each meal should still eat some staple food.” Please rephrase!
The authors’ response:
Thank you for your careful comment. We have changed it to "However, it is essential to include a portion of staple foods in every meal." to make the article more academic.
The reviewers’ comments:
- Please rephrase “Given these pressing health concerns, enhancing the nutritional quality and functional properties of food is a shared priority across various fields such as crop genetics, food science, and nutritional medicine”, to be more clear!
The authors’ response:
Thank you for your careful comments. The original was too broad in scope, and after taking your professional advice, Change to "Given these pressing health concerns, enhancing the nutritional quality and functional properties of rice has become a shared priority across diverse fields, such as rice genetic breeding, scientific processing, and nutritional medicine”. More focused on describing rice, in line with the theme of the full paper.
The reviewers’ comments:
- From introduction and keywords it is not clear that the colored rice; low gluten rice; high resistant starch rice; micronutrient-rich rice; are various types of functional rice or are different types of rice and functional rice is another type of rice! Please clarify!
The authors’ response:
Thanks for the careful comments. This article reviews five different functional rice species, which have five different types of functions. In the position of introduction, we will give a clearer description of these five different functional rice and put them at the beginning of introduction.
The reviewers’ comments:
- Keywords: “bioreactor” or “bioreactor rice”? Are two different concepts!
The authors’ response:
Thank you for your very helpful comments. We have changed bioreactor to bioreactor rice to make the keyword more consistent with the topic of this article.
The reviewers’ comments:
- You mentioned “These elements such as selenium, iron and zinc are essential for the body's physiological function, but too much or too little can cause health problems, so exposure to toxic elements such as arsenic, cadmium, nickel and lead must be minimized [45].” My expectation from a review is to provide these limits (too much or too little) and to discuss about the content of these elements in the chosen materials (here the rice). This would be valuable information.
The authors’ response:
Thank you for your very helpful comments. We add some information about the deficiency of Co, Fe, Mn and other elements, which will cause problems of anemia and growth inhibition, and the excess will cause neurological, hemochromatosis, impaired cognitive and other diseases. It is believed that this information will give readers a better understanding of the harm caused by the deficiency and excess of some trace elements.
The reviewers’ comments:
You mentioned “Bioreactor rice is engineered to produce pharmaceutical proteins and bioactive compounds, making it a promising tool in the field of biomedicine.” This is a potential, interesting information. The readers would like to find out some examples of application. Else this review is maintained at a general discussion.
The authors’ response:
Thanks for the careful comments. We identified vaccines and melatonin as prominent examples that currently illustrate the practical applications of bioreactor rice.
Reviewer 5 Report
Comments and Suggestions for Authors
In this manuscript, the authors review the research progress on the processing and utilization of functional rice, exploring its potential in enhancing human health through the analysis of coloured rice, low-glutelin rice, high-resistant starch rice, micronutrient-rich rice, and bioreactor rice. They also discuss the impact of various processing techniques on the retention of nutritional components in functional rice. However, the current version of the manuscript is not suitable for publication.
The manuscript is poorly written and needs to be modified. Although the topic is interesting, the authors failed to focus on recent findings in the review, for example, there are only five references have been included in the introduction.
The manuscript should include at least two figures that strongly support their review.
Author Response
Many thanks for the valuable suggestions or comments on our manuscript from the reviewers. As suggested, the manuscript has been examined and revised carefully while the textual and typographical errors have been corrected.
Reviewer #5:
The reviewers’ comments:
In this manuscript, the authors review the research progress on the processing and utilization of functional rice, exploring its potential in enhancing human health through the analysis of coloured rice, low-glutelin rice, high-resistant starch rice, micronutrient-rich rice, and bioreactor rice. They also discuss the impact of various processing techniques on the retention of nutritional components in functional rice. However, the current version of the manuscript is not suitable for publication.
The authors’ response:
Thank you for your objective assessment. On the basis of the original article, we will make some content about regionalism more specific. In addition, diagrams and richer references have also been added to the article, hoping to make it more understandable and enjoyable for readers.
The reviewers’ comments:
The manuscript is poorly written and needs to be modified. Although the topic is interesting, the authors failed to focus on recent findings in the review, for example, there are only five references have been included in the introduction.
The authors’ response:
Thank you for your valuable comments. We added several related literatures to make the paper more detailed, and all of them were published in recent years.
The reviewers’ comments:
The manuscript should include at least two figures that strongly support their review.
The authors’ response:
Thanks for the careful comments, we have revised it.
Round 2
Reviewer 1 Report
Comments and Suggestions for Authors
The authors responded to each made suggestion. The graphical and visual aid added to the work was good. Well done.
Reviewer 4 Report
Comments and Suggestions for Authors
Dear Authors,
Thank you for considering my suggestions. I'm glad that you introduced two figures. Now, with the new title and the mention of "narrative review" your approach is consistent with the declared intention. Therefore in my opinion the manuscript can be published.
Reviewer 5 Report
Comments and Suggestions for Authors
The authors addressed all of my comments correctly and happy to accept them.